# Effect of Fractal Ceramic Structure on Mechanical Properties of Alumina Ceramic–Aluminum Composites

**DOI:** 10.3390/ma16062296

**Published:** 2023-03-13

**Authors:** Xianjun Zeng, Qiang Jing, Jianwei Sun, Jinyong Zhang

**Affiliations:** 1State Key Laboratory of Advanced Technology for Materials Synthesis and Processing, Wuhan University of Technology, Wuhan 430070, China; 13657225706@163.com (X.Z.);; 2Hubei Longzhong Laboratory, Xiangyang 441000, China

**Keywords:** 3D printing, fractal structure, mechanical property

## Abstract

In conventional ceramic–metal matrix composites, with the addition of the ceramic phase, although it can significantly improve the performance of the material in one aspect, it tends to weaken some of the excellent properties of the metal matrix as well. In order to meet the high toughness and high strength requirements of composites for practical production applications, researchers have searched for possible reinforcing structures from nature. They found that fractal structures, which are widely found in nature, have the potential to improve the mechanical properties of materials. However, it is often not feasible to manufacture these geometric structures using conventional processes. In this study, alumina ceramic fractal structures were prepared by 3D printing technology, and aluminum composites containing fractal ceramic structures were fabricated by spark plasma sintering technology. We have studied the effect of the fractal structure of alumina ceramics on the mechanical properties of composites. The compression strength of samples was measured by a universal testing machine and the torsional properties of samples were measured by a torsional modulus meter. The results show that a fractal structure improves the compressive strength of aluminum/alumina ceramic composites by 10.97% and the torsional properties by 17.45%. The results of the study will provide a new method for improving the mechanical properties of materials.

## 1. Introduction

Ceramic-reinforced metal matrix composites integrate many excellent characteristics of metal and ceramic materials. While maintaining the good plastic toughness of metal materials, they have the high hardness, high strength, good wear resistance, and corrosion resistance of ceramic materials [1,2,3]. Ceramic-reinforced aluminum matrix composites are the most representative material of ceramic-reinforced metal matrix composites. A ceramic-reinforced aluminum matrix composite has the characteristics of high specific strength, specific stiffness, elastic modulus, wear resistance, and good dimensional stability [4]. It has great practicability and a broad application prospect in aerospace, automobile manufacturing, precision instruments, electronic packaging, sports equipment, and other fields [5]. However, in practical studies, many researchers have found that with the addition of the ceramic phase, the overall hardness and strength of the composite material has increased, while some of the excellent properties of the original metal matrix, such as toughness and plasticity, have decreased [6]. In order to meet the requirements of high strength, high toughness, high hardness, and excellent torsional resistance of composite materials in practical applications, we need to consider the structure of ceramic materials and composite material fabrication. By observing a variety of phenomena in nature, researchers have found some patterns (Figure 1). For example, mature trees are often able to resist the strong torsional forces caused by strong winds, the threaded structure of shells can make them significantly more resistant to compression, and honeycombs can have huge space inside while still having some mechanical strength [7,8,9]. Through further study of these examples, it is believed that some special and ordered structures in trees and shells are the main reason why they have such excellent mechanical properties. This special and ordered structure has been referred to as the fractal structure (Figure 1).

Fractal is usually defined as “a rough or fragmented geometric shape that can be divided into several parts, each of which is a reduced shape of the whole”, i.e., it has the property of self-similarity. The term fractal was originally used by mathematician Benoit Mandelbrot in 1975, to denote a series of objects with specific characteristics, such as self-similar structure and shape geometry, on all magnification scales [10]. In the past decade, people’s interest in using fractal structures to explore the improvement of mechanical and mechanical properties has greatly increased. According to the definition of fractal structures, we know that self-similarity and iterative generation are two important features of fractal structures. Examples of the results of studies on the effects of fractal structure include, the interlocking properties of hierarchical fractal structures providing better load distribution and energy absorption. Farina et al. [11] analyzed the bending behavior of cementitious composites reinforced by straight fractal titanium alloy rods, and the interlocking mechanism between the fractal rods and the matrix resulted in a 152% increase in bending strength. Another example of the application of fractal structures is the optimization of fluid distribution in tree channels. Wang et al. [12] showed that applying fractal structures to heat exchanger channels can improve the heat transfer coefficient and reduce the pressure drop, compared to conventional heat exchangers. Research on the impact resistance of tree-shaped fractal structures is more in-depth. San Ha Ngoc et al. [13] studied the kinetic energy absorption capacity of thin-walled tubes with bionic tree-like cross-sections. Other projects have developed impact-resistant protection devices based on fractal structures, such as fractal honeycomb, Koch curve, side fractal shapes, and Sierpinski shapes [14,15].

Although these properties have been demonstrated, current applications are limited by the fact that these structures, especially in three-dimensional geometries, are difficult to produce, or often impossible to achieve, with conventional manufacturing techniques. Over the past few decades, the development of 3D printing processes has opened up a number of design opportunities that offer more possibilities for manufacturing such complex structures. Stereolithography appearance (SLA) is one of the most widely used 3D printing techniques [16]. Marco Viccica et al. used SLA technology to fabricate a 3D cross-base fractal structure for shock absorption, and studied the mechanical properties of a 3D Greek cross fractal [17]. They conducted numerical studies on the mechanical behavior of the structure under quasi-static and dynamic compression loads, established a material model, and verified the correctness of the model through experiments. It was found that the energy absorption effect of the three-dimensional cross-base fractal structure was 77% higher than that of traditional foam. Wu et al. [18] proposed a new energy-absorbing protective structure, developed by using SLA technology. They established triangular, square, and pentagonal tree fractal structures, and studied their mechanical behavior and deformation process by quasi-static axial fracture tests. These showed that, compared with a single-wall structure, the tree-shaped fractal structure has greater potential to improve the specific energy absorption and resist impact forces. The tree-shaped fractal design promotes the deformation stability of the thin-wall structure. Compared with the single-wall structure, the energy absorption efficiency and load stability of the tree-shaped structure are improved. The tree fractal design improves the deformation stability of single-walled structures by controlling the geometry and material distribution of the structures.

In order to better explore the effect of fractal structure on the mechanical properties of materials, four kinds of alumina ceramic/aluminum composites, containing a fractal structure, were designed and fabricated. The compression strength of four kinds of samples was measured by a universal testing machine. The torsional properties of the different samples were measured by using a torsional modulus meter. The influence mechanism of the fractal structure was also explored and analyzed.

## 2. Structure Description

### Sierpinski Fractal Structure

The Sierpinski polygon, a self-similar structure discovered by Waclaw Sierpinski in the early 20th century, is one of the most commonly used fractal patterns [19]. However, the mechanical properties of this Sierpinski fractal pattern have not been considered, which has prompted the present study to propose advanced shapes, combined with a Sierpinski fractal, to make composite materials.

Triangle Sierpinski fractals and square Sierpinski fractals are two classical fractals in the category of Sierpinski fractals. Their formation processes are shown in Figure 2a,b, respectively.

Among them, the formation process of the triangle Sierpinski fractal is as follows: the initial structure is an equilateral triangle. In the first step, the initial structure is divided into four identical smaller equilateral triangles, and in the second step, the middle is removed, and a first-order Sierpinski fractal structure is generated, as shown in Figure 2a. For the remaining three small equilateral triangles in the above figure, the above two steps are repeated. The results of this, as shown in Figure 2a, are second-order Sierpinski fractals. The fractal structure of other orders can be obtained in accordance with the above two steps.

The formation process of a quadrilateral Sierpinski fractal is as follows: the initial figure is a square structure. In the first step, the initial square is divided into nine identical smaller squares, and in the second step, a small square in the middle is removed, to generate a first-order Sierpinski fractal structure, as shown in Figure 2b. For the remaining eight small squares, the structure obtained by repeating the operation described above is shown in Figure 2b. Other structures of different orders can be obtained by repeating this process.

In this study, we designed four Sierpinski fractals: triangle (Sier 3), square (Sier 4), pentagon (Sier 5), and hexagon (Sier 6). The structure of each polygon follows the fractal law of Sierpinski. Figure 3 shows the schematic diagram of the four fractal design models. The radius of the outer circle of the largest polygon is 50 mm.

## 3. Experimental Procedure

Regarding the design preparation and testing of alumina ceramic/aluminum composites, the experiments performed can be broadly divided into the following steps.

### 3.1. 3D Printing Experiment

(1)A fractal structure was designed, using the Solidwork modeling software (Siemens AG, GER), and imported into a 3D printer.(2)The CeraBuilder160Pro ceramic laser 3D printer (Hubei Wuhan ILaser Inc., China) and ceramic paste (iLaser Inc., CHN) were used, to fabricate the fractal structure alumina ceramics. The specific printing parameters were as follows: the thickness of the printing layer was 0.1 mm, the size of the laser spot was 140 μm, and the working temperature was 25 °C. Then, the 3D printed-Al_2_O_3_ ceramics were put into an air atmosphere box-type furnace (ksl1700x, Hefei Kejing Materials Technology Co., Ltd., China) to remove photosensitive resin. The steps to remove the photosensitive resin were: first, the temperature was heated from room temperature to 300 °C at a rate of 1 °C/min and then kept stable for 120 min. Second, the temperature was heated from 300 °C to 550 °C at a rate of 0.5 °C/min and then kept stable for 120 min. Finally, the temperature was heated from 550 °C to 800 °C at a rate of 2 °C/min, then kept it at 800 °C for 90 min, and then cooled to room temperature naturally.(3)After removal of the photosensitive resin, the sample was sintered in a vacuum sintering furnace, first at 3 °C/min to 1250 °C, held for 60 min, then at 2 °C/min to 1600 °C, held for 90 min, then cooled at 2 °C/min. After cooling to 300 °C in the furnace, the fractal Al_2_O_3_ ceramic structure was fabricated (Figure 4a). The entire construction process is shown in Figure 5.

### 3.2. Fabrication of Alumina Ceramic/Aluminum Composites

Powder metallurgy (PM) is a flexible technology for manufacturing near-clean shape products. Traditional powder metallurgy usually involves three main steps: mixing of the metal powder with a reinforcing agent, compaction, and sintering at high temperature [20]. However, due to the absence of local heating, conventional PM products have high porosity, which further reduces the performance of these products [21]. Spark plasma sintering (SPS) is an effective method for preparing composite materials [22]. Short time densification and high local temperature limit undesired grain growth during SPS [23]. In addition, the alumina layer on the initial particles is removed by spark application, which significantly eliminates the porosity of the manufactured product [24]. Therefore, in this study, we chose to use SPS to make the alumina ceramic/aluminum composite materials.

Commercial spherical aluminum powder (Shanghai Hushi, purity ~99.7%, particle size ~38 μm) was used as the starting material. Table 1 showed the parameters of the raw materials used in this experiment. Table 2 shows the chemical composition of the aluminum powder. The experiment was divided into the following steps. First, 25 g pure aluminum powder and alumina ceramic structure were charged and compacted in an SPS graphite mold, at room temperature and pressure, and then sintered in an SPS sintering furnace. Sintering was carried out in a vacuum atmosphere, with a heating rate of 50 °C/min and a pressure of 45 MPa. Finally, the sintering temperature reached 550 °C. After reaching 550 °C, the sintering temperature was held constant for 10 min. When the insulation was completed, the sample was cooled to room temperature, under a pressure of 45 MPa, for 30 min. After the above steps, a disc-shaped sample, with a diameter of 50 mm and a thickness of about 6 mm, was produced (Figure 4).

In order to compare the performance difference between an aluminum matrix composite made by adding a fractal structure and traditional methods, we fabricated composites with the same size and weight under the same sintering system. First, we used a balance to measure the weight of each fractal structure, and then weighed the alumina ceramic powder equal to the weight of the fractal structure. Secondly, we mixed alumina ceramic powder and 25 g aluminum powder evenly, by the ball milling method. The uniform powder was charged and compacted in an SPS graphite mold, at room temperature and pressure, and then sintered in an SPS sintering furnace. The sintering system was the same as above.

### 3.3. Test Means

Using an MSA324S-000-DU balance, the density of each sample, containing a different fractal skeleton, was measured by the Archimedes drainage method. The thickness of each sample before and after sintering was measured by Vernier caliper. In order to observe the microstructure of the aluminum matrix composite with a fractal structure, the middle part of the sintered sample was selected and cut into strips with dimensions of 3 mm × 4.5 mm × 36 mm. The cross-section of the sintered aluminum matrix composite was ground with sandpaper, and polished with diamond suspension after grinding. A scanning electron microscope (SEM, Hitachi 3400, Japan) was used to observe the fracture and surface morphology of each sample. The compression properties of all samples were tested on an electronic universal testing machine (MTS810, MTS, USA, loading rate 0.5 mm/min), and the displacement load curve of each sample was recorded. A micrometer was used to measure the lateral displacement of the sample and we calculated Poisson’s ratio. The elastic modulus of the sample was measured by an elastic modulus meter (Grindosonic, Belgium). The shear frequency and torsional frequency of the sample under impact were measured by the torsional modulus instrument (Grindosonic, Belgium), and the torsional modulus of the sample was obtained by calculation. Figure 6 shows the test instruments.

## 4. Results and Discussion

### 4.1. Density

The sintering pictures of composite discs with different fractal structures are shown in Figure 4b. It can be observed that, no matter what kind of fractal structure is used, the composite keeps the disc shape. Figure 7 shows the density of the sintered aluminum matrix composites with different fractal structures. For comparison, a pure aluminum disk was made by SPS in the same way, and the density of the aluminum was known (2.74 g/cm^3^) [25]. We compared the densities of the different samples and found that the density of the aluminum alloy without a fractal structure was 2.7292 g/cm^3^, which is close to the theoretical density, indicating that the sintering regime is reliable, while the densities of the samples with fractal structures were higher than that without a fractal structure, which is due to the fact that the density of alumina ceramic (3.62 g/cm^3^) is greater than that of pure aluminum (2.73 g/cm^3^). The differences between the densities of the different samples, are due to the mass of the fractal structure used. We calculated the theoretical density of each sample based on the mass fraction of alumina ceramics, and found that the actual densities were above 98% of the theoretical density values, indicating that the fabricated composites were densified and reliable in performance.

### 4.2. Microstructure

Figure 8 shows SEM images of the surface and cutting surface of the fabricated alumina ceramic/aluminum composite. Figure 8a shows an SEM image of the surface of the pure aluminum phase. The white substance is the lumpy precipitation phase of aluminum powder when sintering. The production of the precipitated phase is related to the particle deformation caused by the pressure and heating temperature. The pressure in the sintering process causes a small amount of Fe, Cu, Mg, and other elements in the aluminum powder, to precipitate out. When heated, these elements have a high melting point and do not easily form a liquid phase. The black part is a pore that is not closed by sintering shrinkage. Figure 8b–d shows SEM images of the cutting surface of the composite material, showing the interface between the Al_2_O_3_ ceramic and the aluminum matrix, observed at different magnifications. Figure 8b shows the interface phase at a resolution ratio of 400 μm. It can be observed that there is no porosity and impurity arrangement on the interface of the alumina ceramics, and the structure is tight and the porosity is low. Figure 8c,d shows the interface at higher resolutions. At these resolutions, we can observe that ceramic particles are distributed in the 3D printed ceramic lattice, with a maximum particle size of about 20μm. After sintering, the porous ceramic lattice has a rough and inhomogeneous lattice organization due to the presence of a few pores. At the same time, a large number of protrusions and depressions are generated, due to the cracking of the photosensitive resin filled between the particles at high temperatures, which is responsible for the uneven surface of the ceramic phase. It is known that at 800 K, since Mg is more reactive than Al, Mg will react with O in Al_2_O_3_ to form MgO, therefore, the Mg atoms in the aluminum alloy matrix will react with Al_2_O_3_ at the interface [26]:3Mg + Al_2_O_3_→ 3MgO + 2Al

Mg atoms in the aluminum matrix are continuously dispersed to the interface and ceramic phase, and the O atoms of Al_2_O_3_ are dispersed to the aluminum matrix. Therefore, the reaction products at the interface mainly contain Al_2_O_3_, MgO, and Al. The interfacial reaction, and the generation of new substances, will produce a transition layer between the alumina ceramics and the aluminum metal matrix, which makes the ceramic phase and metal closely bonded. Therefore, we observed that there is no gap at the interface between the ceramic phase and the aluminum matrix, and there is not a large number of pores and impurities.

Figure 9a shows an SEM image of the cutting surface at a resolution of 10 μm. Figure 9b–d shows the EDS distribution of the elements Al, O, and Mg, respectively. It can be seen from the element distribution diagram that Mg is enriched on the aluminum matrix side, while the content is less on the alumina ceramic side. The distribution of O is more toward the alumina ceramic side and less on the aluminum matrix side. This shows that Mg in the aluminum matrix disperses continuously to the interface and ceramic phase, while O atoms of Al_2_O_3_ disperse to the aluminum matrix, and the existence of an interface transition layer during sintering.

### 4.3. Compression Strength

In this experiment, we put the pure aluminum and aluminum matrix composite discs under the universal testing machine, and recorded the displacement–load curves of the samples. At the same time, the elastic modulus of each sample was measured, by an elastic modulus meter. Since too much load will make the deformation of the sample enter the stage where elastic deformation and plastic deformation act together, the maximum load applied to each sample was 90 KN.

Figure 10a shows the displacement load curves of the pure aluminum sample and samples with fractal structures, and the maximum load applied to each sample is 90 KN. As can be seen from the figure, the fluctuation of the displacement–load curve is relatively small and the curve is smooth, indicating that the whole sample has no structural damage during the compression test. For aluminum matrix composites with powders as structural units, the mechanical properties depend mainly on the bonding state between the powders. The results show that the bond between the powders in each sample is good. Table 3 shows the maximum displacement of each sample under the maximum load, and comparing the relative density of each sample and the mass fraction of Al_2_O_3_ ceramics.

Figure 10b shows the displacement–load curves for the samples containing the four fractal structures, as well as the pure aluminum sample, under high loads (80–90 MPa). Figure 10c shows the relationship between the Poisson’s ratio of the sample and the mass fraction of the ceramics in the sample. It can be seen from the table and figure that the addition of a fractal structure reduces the displacement of the composite under the maximum load to varying degrees, among which, the fractal structure of Sier 6 has the greatest influence, reducing the maximum displacement by 0.16124 mm, while the fractal structure of Sier 3 has the least influence, where the maximum displacement is only reduced by 0.07314 mm. With an increase in the ceramic content in the sample, the maximum displacement of the sample under high load gradually decreased with the use of different fractal structures, and the reduction ratio of displacement was between 6.15% and 13.56%. We hypothesize that the improvement in the compressive strength of the sample was positively correlated with the ceramic mass fraction of the alumina containing ceramics. To verify our conjecture, we tested the elastic modulus of each sample and plotted the relationship between the elastic modulus and the ceramic mass fraction (Figure 10d).

As can be seen from Figure 10d, with an increase in the mass fraction of alumina ceramics in the sample, the elastic modulus of the sample also increases, reaching a maximum of 341.48 GPa, with the increase of the elastic modulus ranging between 5.04 and 10.97%, indicating that the addition of an alumina ceramic fractal structure improves the compressive performance of the composite. The influence of each fractal structure on the compressive resistance is consistent with the mass proportion of the structure. The mass fraction of alumina ceramic in the Sier 3 structure is 17.98%, and its increase in the elastic modulus of the sample is 5.04%. The mass fraction of the alumina ceramic in the Sier 4 structure is 24.57%, and the increase in the elastic modulus of the sample is 8.13%. The mass fraction of alumina ceramic in Sier 5 structure is 32.63%, and its increase in the elastic modulus of the sample is 10.65%. The mass fraction of alumina ceramic in the Sier 6 structure is 33.75%, and its increase in the elastic modulus of the sample is 10.97%. According to previous studies, Zamani et al. [27] prepared nano-Al_2_O_3_ particle-reinforced aluminum matrix composites by a traditional powder metallurgy method, and analyzed the effects of the Al_2_O_3_ content on the microstructure and mechanical properties of the aluminum matrix composites. The results showed that the hardness and compressive strength of the composites were improved with the increase in Al_2_O_3_ content. Nassar et al. [28] studied the wear and mechanical properties of aluminum matrix composites reinforced by nano-TiO_2_ particles with different contents, and obtained similar strengthening laws. The results of the above studies suggest that our result is reliable.

Through the previous part of research and literature reading, we concluded that the main reasons for the improvement in the compressive strength of composite materials containing an alumina ceramic fractal structure are as follows:

(1)The addition of a fractal structure of alumina ceramics reduces the defects in the aluminum matrix. As can be seen from Figure 8a, there are defects such as pores and precipitates on the surface of the aluminum matrix, while alumina ceramics have fewer internal defects, due to solid sintering. After the ceramic structure is added, the alumina ceramic is evenly distributed in the aluminum alloy matrix, which effectively reduces the porosity and defects in the sample (Figure 11), hinders the plastic deformation of the aluminum alloy matrix, and is conducive to the improvement of the compressive strength.(2)Al_2_O_3_ in the ceramic matrix reacts with Al and Mg in the aluminum matrix to form a transition layer. The transition layer connects the aluminum alloy matrix to the ceramic structure, enhances the interface bonding, and promotes the load transition between the two. The robust Al_2_O_3_–Al interface can effectively carry out load transfer, thus delaying the occurrence of interface depolymerization. When the matrix is under pressure, Al_2_O_3_ will play the role of crack bridging, due to the strong bonding of the Al–Al_2_O_3_ interface. Not only that, but the transition layer acts as a dense spherical shell that protects the ceramic structure from damage. Therefore, the ceramic fractal structure can maintain the structural integrity under large compressive loads and further hinder the occurrence of displacement.

The fractal ceramic structures used in this study have ordered structures, while the distribution of ceramic phases in traditional ceramic reinforced composites is homogeneous and disordered. In order to compare the effects of ordered and disordered structures on the mechanical properties of composites, we compared the elastic modulus of alumina ceramic/aluminum composites prepared by adding alumina ceramic powder.

Figure 12 shows the comparison of elastic modulus. The two curves represent the elastic modulus of the alumina ceramic composite reinforced with a fractal structure in this experiment, and the elastic modulus of the alumina ceramic/aluminum composite made with alumina ceramic powder. By comparing the elastic modulus of the composites made by the two methods, with the same ceramic content, we found that the elastic modulus of the composites strengthened with fractal structure ceramics are larger, indicating that their compression performance is better. The reason is, that the fractal ceramic structures used in this paper are ordered structures, while in the composite made of alumina ceramic powder, the ceramic phases are scattered and disordered. Wan et al. [29], for example, made alumina/aluminum composites with a pearl-like microlayer structure. They found that the distribution of oxygen elements in the composite was uniform, which proved that alumina was uniformly distributed in the aluminum matrix. Chen et al. [30] prepared a new type of porous aluminum alloy composite and also found that the distribution of alumina in the composite was uniform. This proves that ceramic phases in composites made of alumina ceramic powders are scattered and disordered.

From the results of the elastic modulus, we can see that the addition of ordered structure has a greater improvement on the compressive properties of composites, than that of disordered structure. This is due to the structural characteristics of ordered structures, and the fact that ordered structures have fewer defects after solid sintering, before making composites.

### 4.4. Torsional Strength

As one of the important mechanical properties of metal and its composite materials, torsional strength has not attracted much attention in the previous research on composite materials. In order to test the influence of ceramic fractal structure on the torsional properties of aluminum, we used the torsional modulus instrument to measure the shear frequency and torsional frequency of the sample under impact force, and then obtained the torsional modulus, *G*, of the sample by calculation. *G* = 116.29 GPa for the pure aluminum sample. Table 4 shows the torsional modulus of each composite and the mass fraction of the fractal structure in each composite.

According to the data obtained, the relationship between the mass fraction of the fractal structure and the torsional modulus of the sample is plotted (Figure 13). It can be seen from Figure 13 and Table 4 that the torsional modulus of the sample increases with the addition of the fractal structure, indicating that the torsional properties of the sample are enhanced, and the torsional modulus of the sample increases from 6.66% to 17.45%. Moreover, with the increase in the mass fraction of the fractal structure contained in the sample, the torsional modulus of the sample also gradually increases.

We know that, one of the most important toughening mechanisms for ceramic/metal composites is the crack bridging of the ductile tube ligament, in which an unbroken metal layer straddles the crack wake and then pulls out [31]. The unfractured metal toughens the composite, by resisting crack opening displacement. When the metal ligament finally breaks, the ceramic begins to pull out of the metal phase, causing frictional sliding, which dissipates the strain energy [32]. In addition, the “multiple cracking” fracture mode also contributes to the cracking resistance of the composite [33]. Multiple cracks reduce crack tip stress, due to greater damage distribution and higher energy absorption, resulting in higher toughness. Le et al. [34] studied the high-cycle fatigue behavior of three kinds of cast aluminum alloys with different microstructure characteristics, under axial, torsional, and proportional tension-torsional loading conditions. Under different loading conditions, the fracture of all failure samples was caused by defects. Under axial loading, the location of initiation defects on the whole section is random, while under torsional loading, the location of initiation defects is near the surface. This is due to the difference in volume at high stresses, under different loading conditions. The entire section is in a high stress state when loaded axially, while there is a stress gradient when loaded torsionally, and only the outer region is in a high stress state [35].

The principle of adding fractal structure ceramics to improve the torsional performance is as follows: with the addition of the parting structure, the porosity of the original aluminum matrix is reduced to a certain extent, the number and distribution of internal defects are improved, and the torsional performance is improved. In addition to the crack mode, the influence of the interface strength is also significant. It is well known that interface strength plays a key role in the toughening of ceramic/metal composites [36]. At the same time, due to the different mass fraction of the fractal structure, the contact area between it and the aluminum matrix is also different. The larger the mass fraction of the fractal structure, the more complete the contact between it and the aluminum matrix. As described in the previous chapter, the interface between the ceramic phase and the aluminum metal creates a layer of transition, which enhances the interface bonding and promotes the load transition between the two. Therefore, the larger the mass fraction of the fractal structure, the stronger the interface bonding. It is known that crack initiation occurs on the surface farthest from the center of rotation, under torsional fatigue [37]. The annular features are formed due to shear stress and abrasion between the two separated surfaces. These annular features converge towards the central region of rotation, where the final rupture occurs. In the final fracture zone, equiaxial dimples appear, and they are formed due to local ductile fracture of the material [38]. Therefore, the existence of the transition layer between the ceramic phase and the metal phase can hinder the crack development and promote the deflection or bridging of the crack [39]. The crack expands in the Al grain and terminates in the adjacent Al_2_O_3_ grain, showing obvious crack passivation characteristics [40]. At the same time, the transition layer may cause crack propagation and deflection, thus consuming more energy [41]. Since crack initiation occurs on the surface farthest from the rotation center, the transition layer of a fractal structure with a larger mass fraction is in contact with the crack earlier than that of a fractal structure with a smaller mass fraction, that is, it has a greater impact on crack growth and load transfer. This explains why the addition of a fractal structure with a larger mass fraction, can lead to a greater improvement in the torsional properties of the sample.

We also calculate the torsional modulus of the composite with a homogeneous ceramic phase at the same ceramic content. Both groups of curves tend to be linear, indicating that the addition of the ceramic phase uniformly enhances the torsional properties of the composites. Under the same ceramic content, the torsional modulus of the composite with a fractal ceramic structure is greater than that of the composite with a homogeneous ceramic phase. The results show that the fractal structure is more effective than the homogeneous ceramic phase in enhancing the torsional properties of composites, and the ordered structure is more effective than the disordered structure in enhancing the torsional properties of composites.

By comparing the effects of fractal structures on the elastic modulus and torsional modulus, it is found that the influence of fractal structures on the torsional properties of composites is greater than that on their compressive properties, when the composites contain the same fractal structures.

## 5. Conclusions

In this paper, four kinds of fractal structures: triangle, square, pentagon, and hexagon, were designed and fabricated. Using SPS technology, the ceramic fractal structures and aluminum powder were fabricated into aluminum matrix composites, with certain mechanical properties. The following conclusions were obtained by testing various mechanical properties of the ceramic fractal structures.

(1)The results of SEM and elemental analyses show that the addition of a fractal structure reduces the defects of the aluminum matrix, and the interface reaction produced by sintering will produce a transition layer between the alumina ceramic and aluminum matrix, so that the ceramic phase and metal bond closely.(2)The addition of a ceramic fractal structure can improve the compressive and torsional properties of composite materials. The increase range of the elastic modulus is 5.04–10.97%; the increase in the torsion modulus is 10.65–34.97%.(3)A fractal ordered structure enhances the mechanical properties of composites more than a homogeneous structure. When the composite materials contain the same fractal structure, the influence of the fractal structure on the torsional properties is greater than that on the compression properties.

## Figures and Tables

**Figure 1 materials-16-02296-f001:**
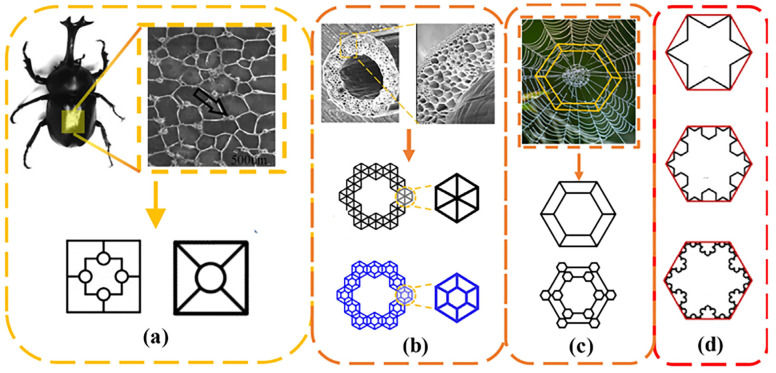
Bio-inspired structures and fractal structures. (**a**) Insect-inspired fractal structure, (**b**) the presence of fractal structures in wood, (**c**) spider’s web of fractal structure, (**d**) the evolution of fractal structures.

**Figure 2 materials-16-02296-f002:**
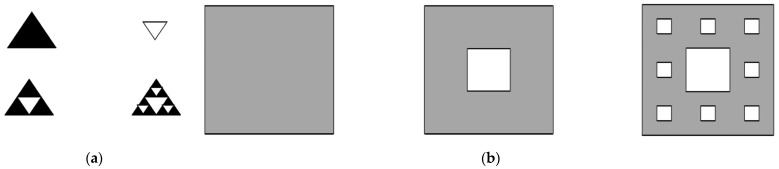
(**a**) Triangle Sierpinski fractal, (**b**) square Sierpinski fractal.

**Figure 3 materials-16-02296-f003:**
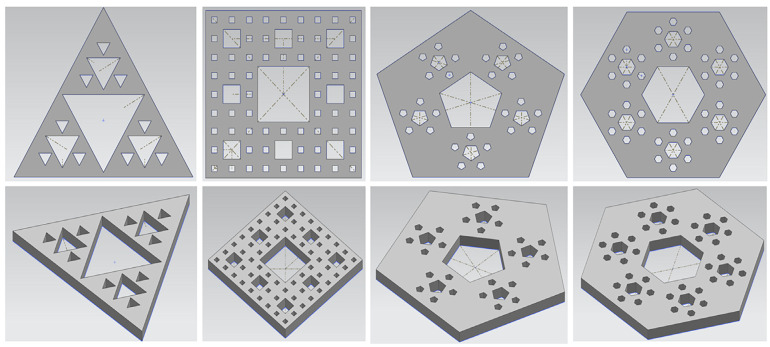
Sierpinski fractal structure model diagram, from left to right, triangle, square, pentagon, and hexagon.

**Figure 4 materials-16-02296-f004:**
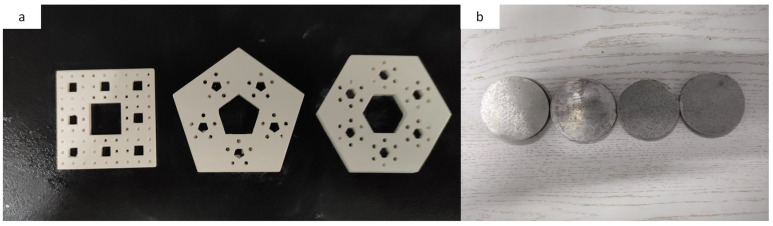
(**a**)The appearance of fractal structure (**b**)composite disk made with SPS.

**Figure 5 materials-16-02296-f005:**
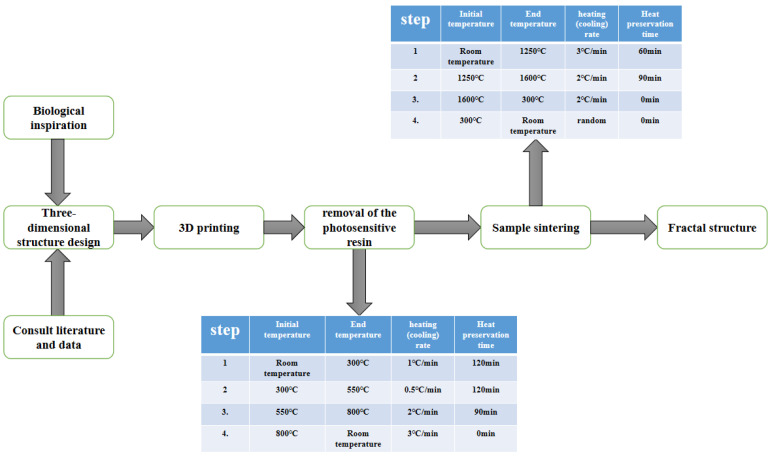
The process of making the fractal structure.

**Figure 6 materials-16-02296-f006:**
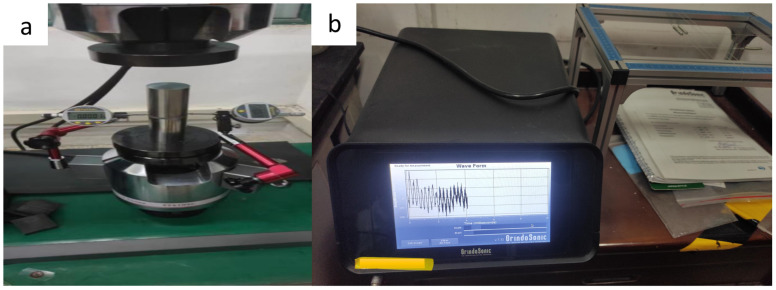
(**a**) Universal testing machine and micrometer, (**b**) elastic modulus instrument.

**Figure 7 materials-16-02296-f007:**
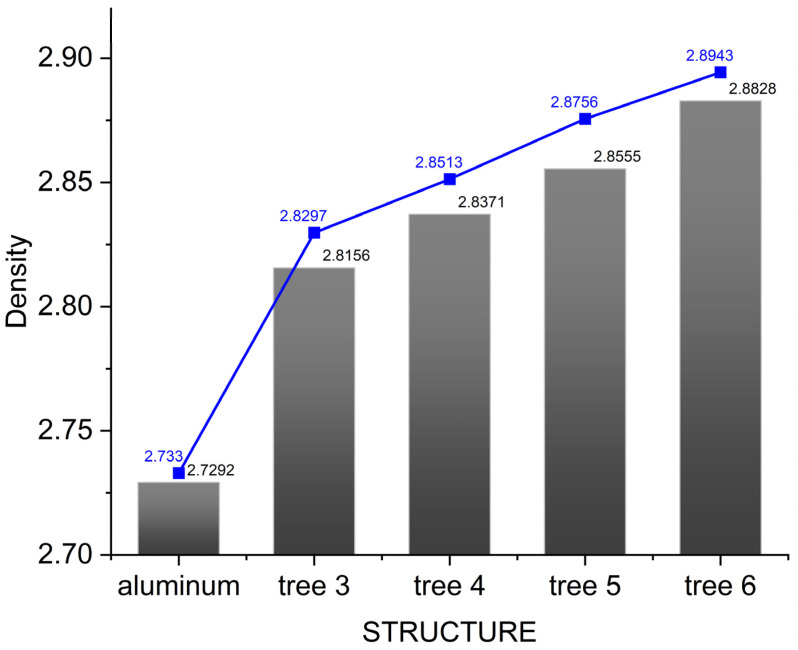
The density of the sintered aluminum alloy composites with different fractal structures. The blue line represents the theoretical density of the composite.

**Figure 8 materials-16-02296-f008:**
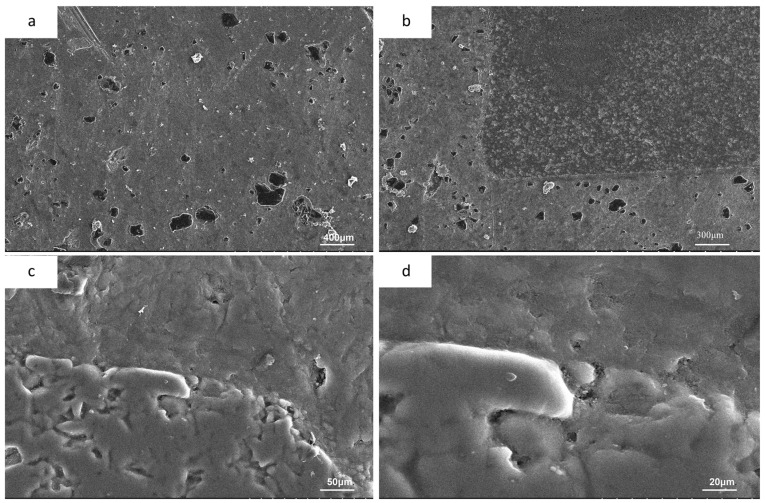
(**a**) The surface of pure aluminum phase, (**b**) the interface phase at 400 μm resolution ratio (**c**) interface phase at 50 μm resolution, ratio(**d**) interface phase at 20 μm resolution ratio.

**Figure 9 materials-16-02296-f009:**
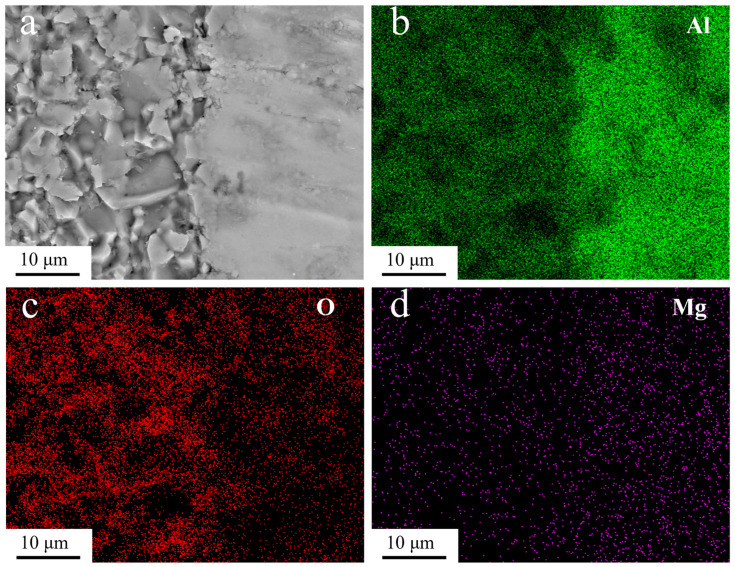
(**a**) SEM images of cutting surface at 10 μm resolution, (**b**) aluminum distribution map, (**c**) oxygen distribution diagram, (**d**) magnesium distribution map.

**Figure 10 materials-16-02296-f010:**
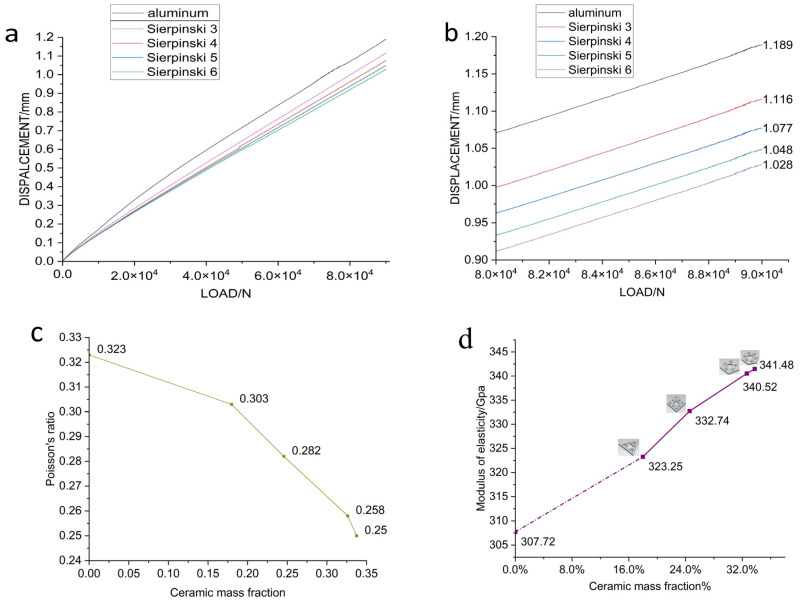
(**a**) Displacement–load curves of samples, (**b**) displacement–load curves of the disks with the four fractal structures and the pure aluminum disk, under high load (80–90 MPa), (**c**) relationship between the Poisson’s ratio of the sample and the mass fraction of the ceramics in the sample, (**d**) relationship between the elastic modulus and the mass fraction of the ceramics in the sample.

**Figure 11 materials-16-02296-f011:**
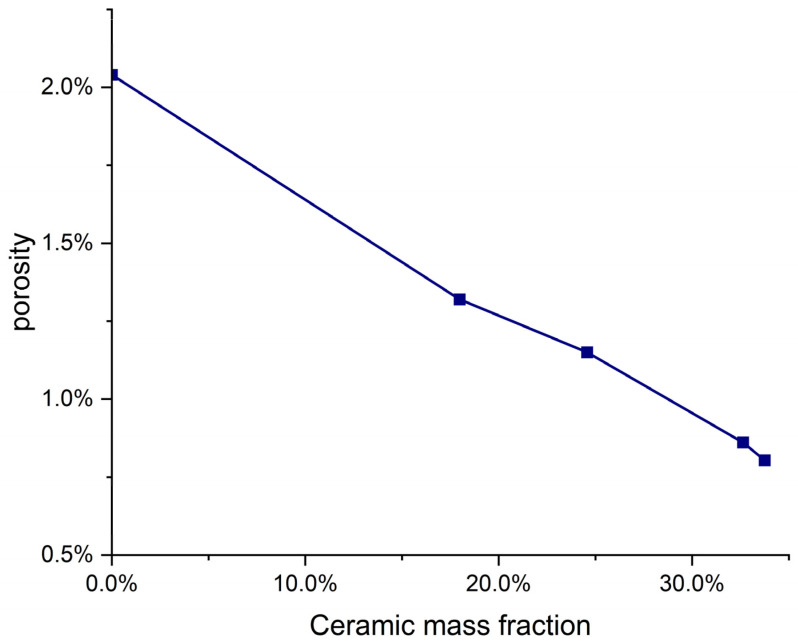
The relationship between porosity and mass fraction of ceramics.

**Figure 12 materials-16-02296-f012:**
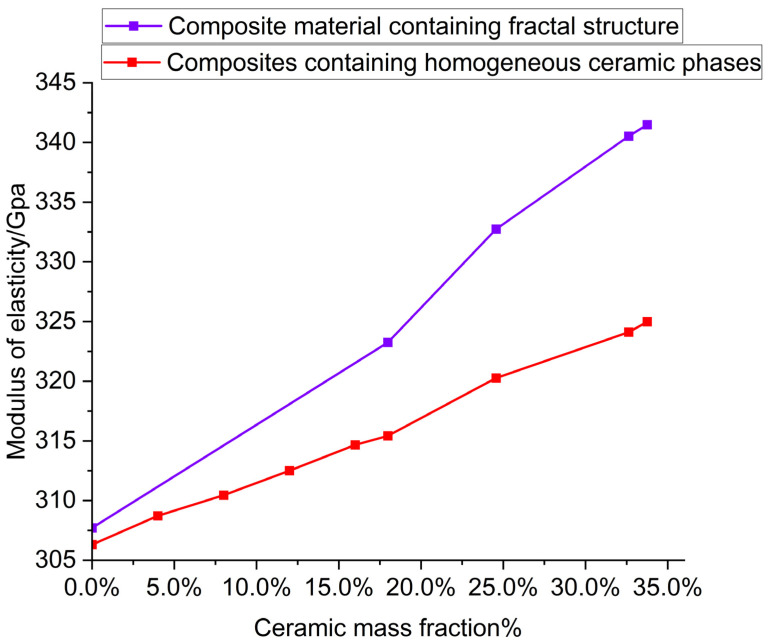
The relationship between elastic modulus and ceramic content of alumina ceramic/aluminum composites made by two different methods.

**Figure 13 materials-16-02296-f013:**
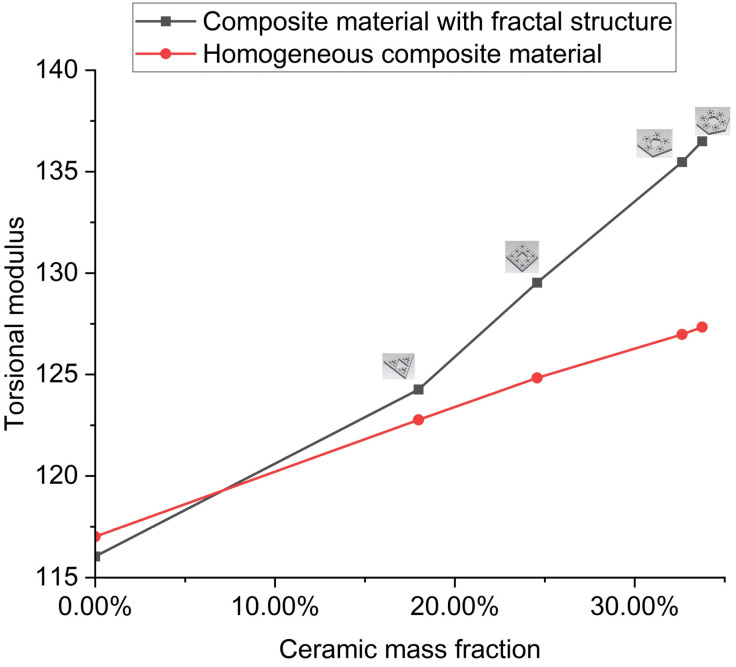
Comparison of torsional modulus.

**Table 1 materials-16-02296-t001:** The parameters of the raw materials used in this experiment.

Powder Type	Particle Size/μm	Purity/%	Manufacturer
Al	38-40	99.7	Shanghai Hushi
Al_2_O_3_	2~3	99.9	Sumitomo chemical company

**Table 2 materials-16-02296-t002:** Chemical composition of the pure aluminum powder used. (Information provided by supplier).

Al	Si	Fe	Cu	Zn	Mn	Mg	Ni
>99.7%	<0.15%	<0.2%	<0.02%	<0.02%	<0.02%	<0.02%	<0.02%

**Table 3 materials-16-02296-t003:** Relationship between maximum displacement and ceramic content.

	Sierpinski				Al
	3	4	5	6	
Density/(g/cm^3^)	2.8504	2.9047	2.9675	2.9763	2.7292
Mass fraction	17.98%	24.57%	32.63%	33.75%	0
Maximum displacement/mm	1.1158	1.0772	1.0483	1.0277	1.18894

**Table 4 materials-16-02296-t004:** Torsional modulus and ceramic content.

		Mass Fraction of Fractal Structure	*G* (GPa)
Sierpinski	3	17.98%	124.26
	4	24.57%	129.53
	5	32.63%	135.47
	6	33.75%	136.50
Al		0	116.03

## Data Availability

Not applicable.

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
