# Peer review of "Effect of Fractal Ceramic Structure on Mechanical Properties of Alumina Ceramic–Aluminum Composites"

_materials, 2023, doi:10.3390/ma16062296_

Round 1
Reviewer 1 Report
In this study, the effect of fractal structure on mechanical properties of alumina ceramics was studied. The compression strength of four samples was measured by universal testing machine, and the torsional properties of different samples were obtained by using the measured data of torque wrench and subsequent calculations. The results show that the fractal structure improves the compressive strength of aluminum/alumina ceramic composites by 10.97% and the torsional properties by17.45%. In this regard, this study will be beneficial to the literature. It can be accepted after minor changes.
1. Language should be reviewed in general. Some places are difficult to understand by the reader.
2. In section 3.1, for better reader understanding, the steps can be given in a chart.
3. In section 3.2., authors said that “In order to compare the performance difference between aluminum alloy composite discs made by adding fractal structure and traditional methods, we fabricated composite discs with the same size and weight under the same sintering system. The only change made is to replace the alumina ceramic fractal structure with an alumina ceramic powder of equal weight.” These change rates and contents should be expressed in tables.
4. In section 3.3, authors should give the pictures of the test setup or test machine.
5. Fig.6, 9, 10, and 11are not clear. These should be given more clearly and visibly.
6. In section 4.3, the data obtained from the test results should be interpreted by supporting the results of the literature.
7. In section 4.3, authors said that “By comparing the elastic modulus of the composites made by the two methods with the same ceramic content, we found that the elastic modulus of the composites strengthened with fractal structure ceramics is larger, indicating that its compression performance is better. The reason is that the fractal ceramic structure used in this paper is an ordered structure, while in the composite made of alumina ceramic powder, the addition of ceramic phases is scattered and disordered.” The result of this should be explained in more detail. should be based on previous studies in the literature.
8. Conclusion is so long. It should be given more specific results and discussions.
9. References should be updated. Especially authors should add the extra new research such as “Influence of replacing cement with waste glass on mechanical properties of concrete” , “Composition Component Influence on Concrete Properties with the Additive of Rubber Tree Seed Shells”, “Influence of replacing cement with waste glass on mechanical properties of concrete”.
Author Response
Response to reviewers
We truly appreciate the reviewer’s comments and suggestions. The manuscript has been revised and the major changes are marked in red in the latest draft. Our response to each comment is included below. We hope we have fully answered all of the reviewer’s questions.
- Language should be reviewed in general. Some places are difficult to understand by the reader.
Response: The English grammar has been improved by professionals to correct grammatical errors and sentence irregularities
- In section 3.1, for better reader understanding, the steps can be given in a chart.
Response: The text description was corrected, and the steps were drawn in Figure 5
- In section 3.2., authors said that “In order to compare the performance difference between aluminum alloy composite discs made by adding fractal structure and traditional methods, we fabricated composite discs with the same size and weight under the same sintering system. The only change made is to replace the alumina ceramic fractal structure with an alumina ceramic powder of equal weight.” These change rates and contents should be expressed in tables.
Response:We improved the text description in this section, described the raw material equipment and manufacturing process needed to make composite materials, and reelaborated the design concept
- In section 3.3, authors should give the pictures of the test setup or test machine.
Response:The pictures of the test setup or test machine has been given in fig. 6.
- 6, 9, 10, and 11are not clear. These should be given more clearly and visibly.
Response:The visual effects of the chart were redesigned to ensure that the information in the graph was clearly visible
- In section 4.3, the data obtained from the test results should be interpreted by supporting the results of the literature.
Response:The supporting literature for Section 4.3 has been supplemented by references [27] and [28] respectively.
- In section 4.3, authors said that “By comparing the elastic modulus of the composites made by the two methods with the same ceramic content, we found that the elastic modulus of the composites strengthened with fractal structure ceramics is larger, indicating that its compression performance is better. The reason is that the fractal ceramic structure used in this paper is an ordered structure, while in the composite made of alumina ceramic powder, the addition of ceramic phases is scattered and disordered.” The result of this should be explained in more detail. should be based on previous studies in the literature.
Response:We rewrote the text of Section 4.3 on elastic modulus and explained the results of comparison in more detail. References [29] and [30] are added to support our argument.
- Conclusion is so long. It should be given more specific results and discussions.
Response:The conclusion section has been rewritten to present the results more concisely and clearly.
- References should be updated. Especially authors should add the extra new research such as “Influence of replacing cement with waste glass on mechanical properties of concrete” , “Composition Component Influence on Concrete Properties with the Additive of Rubber Tree Seed Shells”, “Influence of replacing cement with waste glass on mechanical properties of concrete”.
Response:References have been updated.

Reviewer 2 Report
Article submitted by Xianjun zeng, Qiang Jing, jianwei sun and Jinyong Zhang entitled “Effect of fractal ceramic structure on mechanical properties of alumina ceramic-aluminum composites” highlights the series of tree-shaped alumina ceramic fractal structures were prepared by SLA 3D printing technology, and aluminum composite materials containing fractal ceramic structure were prepared by Spark plasma sintering technology. After quite investigation, I recommend it publication in this journal (major revision) after providing proper improvement in revised version by including suggestion, modification and reply to raised queries which are given below.
Figure 1 is not cited within the text, please double-check the statement to ensure that each figure is mentioned in the manuscript.
The introduction requires more improvement, please clarify the motivation, innovation and contribution of this study. Moreover, more typical references are suggested to be cited in introduction to enrich the various applications of related composites, e.g. 10.1016/j.net.2021.10.007, 10.3390/su15010763
The chemical composition of pure aluminum powder is presented in the study but the ceramic and ceramic composite are not, please provide these chemical analyses.
Investigation of porosity of the nominated composite is very important feature in this study.
Author Response
Response to reviewers
We truly appreciate the reviewer’s comments and suggestions. The manuscript has been revised and the major changes are marked in red in the latest draft. Our response to each comment is included below. We hope we have fully answered all of the reviewer’s questions.
Article submitted by Xianjun zeng, Qiang Jing, jianwei sun and Jinyong Zhang entitled “Effect of fractal ceramic structure on mechanical properties of alumina ceramic-aluminum composites” highlights the series of tree-shaped alumina ceramic fractal structures were prepared by SLA 3D printing technology, and aluminum composite materials containing fractal ceramic structure were prepared by Spark plasma sintering technology. After quite investigation, I recommend it publication in this journal (major revision) after providing proper improvement in revised version by including suggestion, modification and reply to raised queries which are given below.
Figure 1 is not cited within the text, please double-check the statement to ensure that each figure is mentioned in the manuscript.
Response:Add a text description about Figure 1 and highlight the passages related to Figure 1 in the text.
The introduction requires more improvement, please clarify the motivation, innovation and contribution of this study. Moreover, more typical references are suggested to be cited in introduction to enrich the various applications of related composites, e.g. 10.1016/j.net.2021.10.007, 10.3390/su15010763
Response:The text of the introduction has been revised, and the format of the references has been updated.
The chemical composition of pure aluminum powder is presented in the study but the ceramic and ceramic composite are not, please provide these chemical analyses.
Response:A new table is created, which contains the information of raw materials, including particle size, composition, manufacturer, etc.
Investigation of porosity of the nominated composite is very important feature in this study.
Response:Figure 11 is added to show the relationship between porosity and mass fraction of ceramics.

Reviewer 3 Report
The English language of the article requires a serious revision. There are many grammatical and writing errors in the text. The structure of the sentences is such that it is difficult to understand.
Lines 33-35, this sentence needs to be referred.
Line 40 requires a space before the bracket to insert the reference. Please follow this in the whole text.
Figure 1, please add explanation for subfigures (a-d). It is necessary.
Lines 65-66, the reference should be mentioned in the first place referred to in the relevant article.
Lines 75 to 76, it require references.
In my opinion, Section 2 “Structure description” should be a subset of “Experimental procedure” section.
Line 143, please state the name of the manufacturer, city, and country from where the equipment was sourced. Please consider this for all the devices used in this work.
Line 154, what do you mean by “in sintering”? This step is only to remove the resin.
Lines 165 to 176, this paragraph must be transferred to the introduction.
Line 171, there is still a lot of discussion among researchers about the existence or absence of spark discharge or plasma in the SPS. It is recommended to avoid writing it. See this article for more information (A critical review on spark plasma sintering of copper and its alloys).
Lines 180-182, please insert the amount of cold pressure.
Lines 182-183, please use heating rate instead of “temperature rise of”.
lines 187-191, this information is not enough for preparation method of analogous samples. You should extent it and mention the mixing method and reinforcement/matrix ratio and so on.
Table 1, how do you get these data? Is the data that the supplier has provided? Please specify it.
Figure 5 is given without any logic and must be eliminated.
Please add a subdivision under the name of "raw materials" and list all the materials used with the size, purity, and the manufacturer.
Please provide a table and name the production samples along with their specifications.
I don't understand the logic behind Section 4.1. In fact, it does not provide any useful information. Either deleted or must be re -written.
Figure 6, please take more time in the preparation of figures. the visuality of charts is very important.
Author Response
Response to reviewers
We truly appreciate the reviewer’s comments and suggestions. The manuscript has been revised and the major changes are marked in red in the latest draft. Our response to each comment is included below. We hope we have fully answered all of the reviewer’s questions.
The English language of the article requires a serious revision. There are many grammatical and writing errors in the text. The structure of the sentences is such that it is difficult to understand.
Response: The English grammar has been improved by professionals to correct grammatical errors and sentence irregularities
Lines 33-35, this sentence needs to be referred.
Response:Corresponding references have been added
Line 40 requires a space before the bracket to insert the reference. Please follow this in the whole text.
Response:The format has been modified here, and the format of the full text has been corrected
Figure 1, please add explanation for subfigures (a-d). It is necessary.
Response:a description of the subgraphs (a-d) has been added, which are.(a)Insect-inspired fractal structure (b)The presence of fractal structures in wood (c)Spider web of fractal structure (d)The evolution of fractal structures
Lines 65-66, the reference should be mentioned in the first place referred to in the relevant article.
Response:The reference location has been modified。
Lines 75 to 76, it require references.
Response:Corresponding references have been added.
In my opinion, Section 2 “Structure description” should be a subset of “Experimental procedure” section.
Response:Among the references about 3D printing structures, the chapter "structure design" exists independently, because most of them describe the design concept and rejection process of structures, while the chapter "Experimental procedure" describes the preparation of experimental samples, so I will separate the chapter "structure design".
Line 143, please state the name of the manufacturer, city, and country from where the equipment was sourced. Please consider this for all the devices used in this work.
Response:the name of the manufacturer and source has been added.
Line 154, what do you mean by “in sintering”? This step is only to remove the resin.
Response:The relevant text description has been modified to avoid ambiguity.
Lines 165 to 176, this paragraph must be transferred to the introduction.
Response:Lines 165 to 176 simply state the reason for choosing SPS sintering technology, which is not directly related to the topic of this paper, so I did not choose to put it in the introduction.
Line 171, there is still a lot of discussion among researchers about the existence or absence of spark discharge or plasma in the SPS. It is recommended to avoid writing it. See this article for more information (A critical review on spark plasma sintering of copper and its alloys).
Response:The controversial part was cut out.
Lines 180-182, please insert the amount of cold pressure.
Response:The amount of cold pressure has been insert.
Lines 182-183, please use heating rate instead of “temperature rise of”.
Response:The ambiguous statement has been modified.
lines 187-191, this information is not enough for preparation method of analogous samples. You should extent it and mention the mixing method and reinforcement/matrix ratio and so on.
Response:We improved the text description in this section, described the raw material equipment and manufacturing process needed to make composite materials, and reelaborated the design concept.
Table 1, how do you get these data? Is the data that the supplier has provided? Please specify it.
Response:Fixed information in Form 1 (provided by Sponsor)
Figure 5 is given without any logic and must be eliminated.
Response:The original figure 5 has been deleted and replaced with the experimental procedure diagram.
Please add a subdivision under the name of "raw materials" and list all the materials used with the size, purity, and the manufacturer.
Response:Information about raw materials has been added
Please provide a table and name the production samples along with their specifications.
Response:Table added.
I don't understand the logic behind Section 4.1. In fact, it does not provide any useful information. Either deleted or must be re -written.
Response:Section 4.1 mainly describes the relationship between the density of the prepared sample and the theoretical density, in order to prove that the prepared sample is dense. The text of this section has been rewritten to reduce ambiguity.
Figure 6, please take more time in the preparation of figures. the visuality of charts is very important.
Response:The visual effects of the chart were redesigned to ensure that the information in the graph was clearly visible

Round 2
Reviewer 1 Report
revisions were made. it should be accepted.
Reviewer 2 Report
Accepted in current form
Reviewer 3 Report
The authors adequately improved the manuscript and addressed all issues. I can now recommend it for publication in its current form.